# Identifying Candidate Gene Drivers Associated with Relapse in Pediatric T-Cell Acute Lymphoblastic Leukemia Using a Gene Co-Expression Network Approach

**DOI:** 10.3390/cancers16091667

**Published:** 2024-04-25

**Authors:** Anthony Kypraios, Juba Bennour, Véronique Imbert, Léa David, Julien Calvo, Françoise Pflumio, Raphaël Bonnet, Marie Couralet, Virginie Magnone, Kevin Lebrigand, Pascal Barbry, Pierre S. Rohrlich, Jean-François Peyron

**Affiliations:** 1Université Côte d’Azur, Inserm C3M, 06200 Nice, Franceveronique.imbert@unice.fr (V.I.); lea.david@etu.univ-cotedazur.fr (L.D.); raphael.bonnet.06@gmail.com (R.B.); rohrlich.ps@chu-nice.fr (P.S.R.); 2Team#4: “Fundamental to Translational Research on Dysregulated Hematopoiesis—DysHema”, Centre Méditerranéen de Médecine Moléculaire-C3M-Inserm U1065, Bâtiment Universitaire ARCHIMED, 151 Route Saint Antoine de Ginestière, BP 2 3194, CEDEX 3, 06204 Nice, France; 3Université de Paris, Inserm, CEA, 92260 Fontenay-aux-Roses, France; 4Université Côte d’Azur, CNRS, IPMC, 06560 Valbonne, France; couralet@ipmc.cnrs.fr (M.C.); magnone@ipmc.cnrs.fr (V.M.); lebrigand@ipmc.cnrs.fr (K.L.);; 5UCA GenomiX, IPMC, 06560 Valbonne, France; 6CHU de Nice, Hôpital de l’Archet, 06000 Nice, France

**Keywords:** pediatric T-cell acute lymphoblastic leukemia, treatment resistance, relapse, single-cell RNAseq, gene co-expression networks, hub genes, candidate gene drivers

## Abstract

**Simple Summary:**

Non-genetic transcriptomic plasticity plays a pivotal role in cancer cell resistance to treatments. This study aims to elucidate the molecular mechanisms underlying the transcriptomic evolution from diagnosis to relapse in pediatric T-cell acute lymphoblastic leukemia (T-ALL). We conducted single-cell RNA sequencing analysis on paired diagnosis–relapse samples to address this question. Using hdWGCNA, we constructed gene co-expression networks to identify relapse-associated networks, modules, and hub genes, potentially indicative of leukemic drivers. Through combining results from three pairs of patient samples, we identified the most discriminating genes between diagnosis and relapse using sPLS-DA. A Cox analysis revealed their potential to identify patients with lower survival in the AALL0434 cohort. These relapse hub genes are promising as future therapeutic targets or relapse markers.

**Abstract:**

Pediatric T-cell Acute Lymphoblastic Leukemia (T-ALL) relapses are still associated with a dismal outcome, justifying the search for new therapeutic targets and relapse biomarkers. Using single-cell RNA sequencing (scRNAseq) data from three paired samples of pediatric T-ALL at diagnosis and relapse, we first conducted a high-dimensional weighted gene co-expression network analysis (hdWGCNA). This analysis highlighted several gene co-expression networks (GCNs) and identified relapse-associated hub genes, which are considered potential driver genes. Shared relapse-expressed genes were found to be related to antigen presentation (HLA, B2M), cytoskeleton remodeling (TUBB, TUBA1B), translation (ribosomal proteins, EIF1, EEF1B2), immune responses (MIF, EMP3), stress responses (UBC, HSP90AB1/AA1), metabolism (FTH1, NME1/2, ARCL4C), and transcriptional remodeling (NF-κB family genes, FOS-JUN, KLF2, or KLF6). We then utilized sparse partial least squares discriminant analysis to select from a pool of 481 unique leukemic hub genes, which are the genes most discriminant between diagnosis and relapse states (comprising 44, 35, and 31 genes, respectively, for each patient). Applying a Cox regression method to these patient-specific genes, along with transcriptomic and clinical data from the TARGET-ALL AALL0434 cohort, we generated three model gene signatures that efficiently identified relapsed patients within the cohort. Overall, our approach identified new potential relapse-associated genes and proposed three model gene signatures associated with lower survival rates for high-score patients.

## 1. Introduction

Acute lymphoblastic T-cell leukemia (T-ALL) is a serious and aggressive form of leukemia that can affect children and adolescents [1]. It arises from genetic mutations favoring abnormal proliferation of T-cell progenitors and blocking their differentiation at various stages, leading to the accumulation of immature and nonfunctional lymphoblasts in bone marrow and blood [2]. 

T-ALL cases account for 15% of pediatric and 25% of adult cases of acute lymphoblastic leukemia (ALL), and generally have a slightly worse prognosis compared to B-ALL. Unlike B-ALL, targeted therapies and immunotherapies have not yet been proven effective for T-ALL [3]. Although intensive treatments combining corticoids, anti-tumor antibiotics, mitotic inhibitors, and antimetabolites achieve high cure rates (80–85%), T-ALL relapses still have a dismal outcome for 70% of patients due to drug resistance [4]. It is therefore crucial to identify vulnerabilities in resistant T-ALL cells to develop new selective drugs, guided by an understanding of the molecular mechanisms underlying leukemia development and resistance. Genetic profiling has already identified biological subgroups of T-ALL associated with different gene expression programs and clinical outcomes [5].

The advent of single-cell RNA sequencing (scRNAseq) methods allows for the analysis of global gene expression at the level of individual cells, providing unprecedented resolution of cancer mechanisms [6]. scRNAseq offers a comprehensive view of leukemic cellular heterogeneity, highlighting subpopulations with distinct transcriptomic profiles contributing to leukemia genesis and/or chemoresistance. We generated scRNAseq data for paired samples obtained at diagnosis and relapse from three childhood T-ALL cases. Utilizing high-dimensional weighted gene co-expression network analysis (hdWGCNA), we deciphered the biological processes underlying leukemia evolution. WGCNA, a systems biology approach, clusters genes with similar expression patterns into co-expression modules to reveal genes and their associated biological functions. hdWGCNA facilitated the identification of hub genes, the most connected genes within a module, which are considered potential disease drivers [7]. By comparing hub genes between diagnosis and relapsed T-ALL samples, we elucidated mechanisms associated with treatment resistance and identified relapse-specific hub pathways as potential therapeutic targets. Applying various machine learning models to transcriptomic and clinical data from the TARGET AALL0434 pediatric T-ALL cohort [8], we validated three gene signatures. These signatures, using transcriptomic data obtained at initial diagnosis, could assign 85% of the 20 relapsed patients to a low-survival group.

## 2. Materials and Methods

### 2.1. T-ALL Patient’s Samples

Blood and/or bone marrow samples were obtained from Hôpital R. Debré, Hôpital A. Trousseau (Paris, France), and Hôpitaux Civils de Lyon (Lyon, France) and processed as described previously [9]. Informed consent of patients or of relatives was obtained in accordance with the Declaration of Helsinki and the Ethic regulations. The research project was approved by the ethics evaluation committee of Inserm (IORG0003254, FWA00005831). Human T-ALL samples are detailed in Appendix A. 

### 2.2. 10X Genomics Single-Cell Processing, Libraries Preparation, and Sequencing

DMSO frozen cells were quickly thawed, washed, and resuspended in PBS containing 2% BSA. Cell viability was measured using NucGreen/Hoescht (Merck KGaA, Darmstadt, Germany) staining, and only samples with a >80% cell viability were processed. Cells were loaded on a Chromium Controller (10X Genomics, Pleasanton, CA, USA) with a target output of 5000 cells per sample. Reverse transcription, cDNA synthesis/amplification, and library preparation were performed according to the 10X Genomics protocol (Chromium Next GEM Single Cell 5’Reagent Kits v2 with feature barcoding). The scRNA libraries were sequenced on an Illumina NextSeq 2000 (Illumina, San Diego, CA, USA). The QC parameters are presented in Appendix A. The single-cell datasets have been deposited on the Gene Expression Omnibus portal under series number GSE262271.

### 2.3. scRNAseq Data Analysis

Single-cell RNA-seq data were processed using the Seurat (5.0.2) pipeline [10] (https://satijalab.org/seurat/articles/pbmc3k_tutorial, accessed on 20 April 2024) in R (version 4.2.3). Quality control steps included demultiplexing the data and removing doublet cells and empty droplets using the *HTODemux* function with default parameters. Cells were retained based on the following criteria: (i) a percentage of expressed mitochondrial genes below 5% and (ii) outliers of the number of detected genes (>200 for patient M104/M127, >300 for patient M143/M148, >90 for patient M187/M187r). Subsequently, the data were normalized using the Seurat *NormalizeData* function through the “*LogNormalize*” method, using the median count number for each cell as the scale factor. Principal component analysis (PCA) was performed on the top 2000 most variable genes. A UMAP (Uniform Manifold Approximation and Projection) dimensionality reduction was then conducted on the scaled matrix, utilizing the most important components (combination of all variables), defined as follows according to the Jackstraw test [11]: M104/M127: first 28 components, M143/M148: first 26 components, M187/M187r: first 31 components. The single-cell datasets generated for this study were deposited on the Gene Expression Omnibus portal under the accession number GSE262272. The R scripts used in this article can be found at Peyronlab.github.io.

### 2.4. Gene Co-Expression Network Constructions

A gene co-expression network (GCNs) approach can be utilized to unravel the complexity of biological systems by associating genes involved in similar biological functions and comparing different disease stages [12,13], such as diagnosis and relapse of T-ALL in the present study. GCNs were generated using the R package *hdWGCNA (High-Dimensional Weighted Gene Correlation Network Analysis) version_0.3.01* package (https://github.com/smorabit/hdWGCNA, accessed on 20 April 2024)*,* which analyzes co-expression networks in scRNAseq data [14]. Since WGCNA [15] is sensitive to data sparsity, we defined metacells (aggregates of small groups of transcriptomically similar cells) using original scRNAseq data with the *MetacellsByGroups* function of *hdWGCNA,* and normalized the data afterwards with the *NormalizeMetacells* function. Subsequently, we computed the expression matrix defined by metacells and specified the soft-power threshold according to the original WGCNA R package recommendations [7,15]

All standard analyses were conducted according to the official *hdWGCNA* pipeline, which can be found at https://smorabit.github.io/hdWGCNA/articles/basic_tutorial.html, accessed on 20 April 2024.

### 2.5. Module Projection on Healthy T-Cells Dataset

The modules generated for each patient dataset were compared to a GCNs analysis applied to healthy T-cells from the 10X Genomics Peripheral Blood Mononuclear Cells (PBMC) reference dataset (https://cf.10xgenomics.com/samples/cell/pbmc3k/pbmc3k_filtered_gene_bc_matrices.tar.gz, accessed on 20 April 2024) to select leukemia-specific modules using the *ProjectModules* function of *hdWGCNA*.

### 2.6. Module Preservation Statistics

We assessed the combined preservation statistics of Gene Co-expression Networks (GCNs) by conducting 250 random permutations to confirm the presence of modules in healthy T-cells [16]. Additionally, comparing the quality statistics of the modules to their original dataset enabled us to evaluate the overall quality of the modules generated.

These statistics were computed using the *ModulePreservation* function of the *hdWGCNA* package, which provided various types of compound preservation statistics [6], distinguished here by the “.qual” or “.pres” suffixes. The “.qual” suffix measures module quality by considering the dataset it originates from (the reference dataset) and evaluates the closeness of its nodes and its uniqueness compared to other modules in the same network. The “.pres” suffix indicates preservation statistics of a module constructed with the reference dataset in the projection dataset (in this case, healthy T-cells). We set Z-summary thresholds according to the literature [16,17,18]. A Z-summary > 10 suggested strong module preservation, while 2 < Z-summary < 10 indicated weak to moderate preservation, and Z-summary < 2 suggested significant variation between the reference and projected modules. The *MedianRank* statistic was utilized to compare modules of different sizes [17]. We considered all of these statistics in light of the notion that modules not preserved between healthy T-cells and leukemic T-cells (Z-summary < 10) might elucidate T-ALL-specific biological mechanisms [16]. 

### 2.7. Diagnosis–Relapse, Most Discriminating Genes

The top 25 hub genes for each module were selected according to their kME (Module Eigengene-based connectivity) value, which provides a quantitative measure of how closely their gene expression pattern aligns with the overall expression pattern of a module (1st Component). 

Next, we employed a sparse partial least squares discriminant analysis (sPLS-DA) (R MixOmics package) to identify the most discriminant genes between diagnosis and relapse [19]. During sPLS-DA, the contribution of each variable to the discrimination between classes is measured through loadings. Loadings represent the direct regression coefficients between the original variables and the latent components (linear combinations of the original variables) extracted by the algorithm. Variables with the largest loadings have the most influence in determining the components (Appendix A). 

To optimize the discrimination capacity of sPLS-DA, the number of components was chosen using the MixOmics *perf* function with 10 folds and 50 repetitions, and the *tune* function was used to define the number of variables to be considered by components, with 10 folds and 100 repetitions applied to different vectors, assigning each component with different numbers of variables. Here, a vector is defined between a low and a high limit, with a resolution used to narrow the interval between them. On each iteration the resolution is increased based on the output of the *tune* function.

Once each patient’s library had been processed, the genes with the highest discriminative power were extracted. For each loading vector, we recovered the top 10 genes (or all for vectors containing less than 10 genes) with the best loading vector absolute values per component. 

### 2.8. Relationship between Gene Signatures and Survival

We then assessed the significance of these most discriminating genes in T-ALL patient survival. Quantification was performed using Cox regression applied to the TARGET-ALL cohort (n = 249) to investigate the relationship between time to relapse and the expression of the signature genes. To streamline the selection process in our less intricate signature, we employed Akaike Information Criterion (AIC) optimization with backward search, considering all of the genes identified by sPLS-DA. This method allowed us to select the most appropriate predictors, even incorporating variables that may not individually exhibit significant effects, provided they contribute slightly to improving the overall model fit without overly complicating it. We evaluated the significance of these highly distinguishing genes in determining survival among T-ALL patients by employing Cox regression on the TARGET-ALL cohort, which consists of 249 patients. 

## 3. Results

### 3.1. scRNAseq Analysis of Three Paired Diagnosis–Relapse T-ALL Samples

Our analysis workflow is depicted in Figure 1A. Initially, the scRNAseq data from three paired diagnosis–relapse pediatric T-ALL samples (patient characteristics are provided in Appendix A) were analyzed using UMAP (Figure 1B). For libraries one (patient #1, samples M104/M127) and two (patient #2, samples M143/M148), we observed a clear separation of diagnosis vs. relapsed cells, indicating significant transcriptomic differences. Conversely, some degree of overlap was observed between the two samples from library three (patient #3, samples M187/M187r). This overlapping pattern was evident in the global UMAP representation of the six samples (Appendix A), demonstrating both inter- and intra-patient transcriptomic heterogeneity. This heterogeneity was less pronounced for the diagnosis and relapse samples from patient #3. 

### 3.2. Construction of Gene Co-Expression Networks

Next, we applied hdWGCNA to the data from each library to construct and visualize gene co-expression networks (GCNs). From these GCNs, we extracted gene modules corresponding to genes with similar expression trends, which are represented by UMAP (Figure 2). Each module was labeled with the top gene exhibiting the highest kME. A ForceAtlas (version 0.1, https://github.com/analyxcompany/ForceAtlas2, accessed on 20 April 2024) representation displays the relationships between the modules (Appendix A). The hdWGCNA approach highlighted 37 modules (Appendix A), and the differential expression between diagnosis and relapse states is shown in Appendix A. Among them, 29 were observed only in leukemic cells and not in normal T-cells (Appendix A, Appendix A). The distribution of these leukemic modules was as follows: diagnosis: one for patient #1, four for patient #2, eighteen for patient #3; relapse modules: one for patient #1, three for patient #2, and two for patient #3. We observed significant diversity in the modules at relapse between the three patients (Appendix A). Nevertheless, some similarities could be observed. Analysis of the “relapse” modules (Appendix A) shows that patients #1 and #3 both express a module involved in antigen processing and presentation. Genes of the NF-κB family were observed in patient #1. Patient #2 upregulated a module participating in cytoplasmic translation, ribosome biogenesis, initiation and elongation of translation, as well as a module comprising *FOS/FOSB/JUN/JUND* genes. Patient #3 displayed a module with genes (*TUBB*, *TUBA1B*) that could be involved in cytoskeleton remodeling. 

### 3.3. Search for Conserved Relapse-Associated Hub Genes

We compared the gene networks for each library to select the most conserved hub genes. Initially, we observed that genes associated with MHC class I complex (*HLA-A/B*, *beta-2M*) were common to the three patients, along with two Krüppel-like factors: *KLF2/6*. Additionally, ten other genes (*EIF1*, *EEF1B2*, *FTH1*, *UBC*, *EMP3*, *MIF*, *HSP90AB1/AA1*, *NME1/2*, *ARL4C*) were common in at least two patients. 

### 3.4. Identification of Diagnosis–Relapse Discriminating Genes

By constructing modules and hub genes subnetworks, we filtered 36,000 starting features down to only a few hundred involved in different co-expression networks. Selecting potential predictors, we identified 29 modules (see Section 2.3), each containing 25 hub genes, resulting in 481 unique hub genes (Appendix A). Among these, 110 scaled genes defined by the Seurat workflow at the intersection of the three libraries were analyzed through sPLS-DA (Appendix A). This analysis provides an overall view of the ability of genes to best discriminate leukemic cells between relapse and diagnosis, without concern for the multicolinearity of the variables (Figure 3). 

The discrimination between cells from diagnosis or relapse was less pronounced for patient #3 compared to patients #1 and #2. This reduced transcriptomic disparity for patient #3 could be attributed to a shorter diagnosis–relapse interval (M187/M187r: 11 months) compared to patient #1 (M104/127: 19 months) and patient #2 (M143/148: 23 months). A statistical method was employed to highlight the most important genes, distinguishing diagnosis from relapse. It identified 67 unique genes distributed as 44, 35, and 31 highly discriminating genes for patients #1, #2, and #3, respectively. A Gene Ontology (GO) analysis of these genes is presented in Appendix A. Thirteen genes were found in common between the three patients (Figure 4 refers to the different gene lists in Appendix A).

### 3.5. Establishment of Library Gene Signatures and Correlation with Patient’s Survival

To assess whether these three groups of highest discriminating genes could be associated with patient outcomes, a Cox survival regression analysis was conducted using the transcriptomic and clinical data from 249 T-ALL patients from the TARGET AALL0434 pediatric cohort. A stepwise model selection using an AIC criterion was performed, and one survival model was identified for each library. Models 1, 2, and 3 consist of, 11, 7, and 7 genes, respectively (Appendix A).

Subsequently, a scoring system was developed for each model by multiplying the expression level of each gene by its corresponding Cox coefficient (Appendix A). A “high score” (above the median score expression) was associated with a shorter survival for the patient (Figure 5). Each model score could classify 85% (17 patients), 80% (16), and 90% (18) of the 20 relapses of the cohort in the “High-score” group, respectively. 

In these models, the augmented expression of certain genes was associated with an increase in the score for the patients, as follows: *DUSP6*, *HSP90AA1*, *TPX2*, *TUBA1B*, and *ZFP36* (model 1); *NDC80*, *TUBA1B*, and *ZFP36* (model 2); and *DUSP6*, *HSP90AB1*, and *MIF* (model 3). Conversely, an increased expression of the following genes was associated with a lower score: *ACTB*, *HES4*, *HMGB2*, *JUNB*, *PHGDH*, and *TUBB* (model 1); *CD52*, *DUSP2*, *MKI67*, and *TUBB* (model 2); and *CDK2AP2*, *JUN*, *PFN1*, and *RPS6* (model 3) (Appendix A). Our findings consistently revealed *p*-values well below the conventional significance threshold of 0.05 for Kaplan–Meier plots. This demonstrates the highly significant power of these gene signatures in discriminating between patients who experienced relapses and those who did not. Specifically, 60% of relapses were similarly assigned to the three models, with 15% in models 1 and 2, 15% in models 1 and 3 and, finally, 10% in models 2 and 3 (Appendix A). Appendix A display optimal values of variables for sPLS-DA, results of the Cox regression models and Risk parameters of the Kaplan-Meier analysis.

## 4. Discussion

Cancer cell resistance to treatments can arise due to intrinsic genomic instability [20] and non-genetic mechanisms generating transcriptomic plasticity [21,22]. In this study, we employed a systems biology approach, hdWGCNA, to analyze single-cell RNAseq transcriptomic data obtained from three paired diagnosis–relapse cases of pediatric T-ALL, aiming to unravel the mechanisms of relapse. Our focus was on relapse-associated gene networks and hub genes, examining their conservation across the patients. Hub genes, representing the most connected genes within their module, are considered as potential drivers or significant players of relapse. However, further biological experiments are necessary to evaluate their true driver potential. 

By delving into the complexity of leukemic transcriptomes through hdWGCNA, we constructed gene co-expression networks, consisting of functional modules comprising genes with closely intertwined expression profiles associated with relapse vs. diagnosis. We extracted the top 25 hub genes exhibiting the highest connectivity within their module. Analysis of the different relapse modules revealed considerable diversity among the three patients, suggesting that resistance mechanisms may vary widely among individuals. Nonetheless, some commonalities were noted. For example, at relapse, all three patients expressed modules containing genes of the MHC class I complex (*HLA-A/B*, *beta-2M*) involved in antigen presentation, indicating a potential role in immune interference and contributing to resistance. Additionally, patient #1 exhibited modules expressing genes of the NF-κB family, while patient #2 displayed a module comprising *FOS/FOSB/JUN/JUND* genes, highlighting transcriptional plasticity. Patient #3 showed a module with genes potentially involved in cytoskeleton remodeling, as follows: *TUBB*, *TUBA1B*. Overall, the hdWGCNA approach proved valuable in identifying some molecular mechanisms associated with relapse. To further explore common relapse-dysregulated genes, we searched for “relapse” hub genes shared by all three patients. Aside from *HLA-A/B* and *beta-2M* genes, two Krüppel-like factors, either *KLF2* or *KLF6*, were found, respectively, in samples from patient #3 and patients #1 and #2. 

The zinc-finger transcription factors of the KLF family play crucial roles in various physiological and pathological processes, particularly during development. While *KLF2* and *KLF6* are implicated in T-cell biology, their exact roles remain largely undefined [23]. *KLF2* expression rapidly extinguished after T-cell activation may promote quiescence, survival, and migration, whereas *KLF6* has dual roles depending on the cellular context. *KLF6* acts as a tumor suppressor in various solid cancers (colorectal, prostate, glioma) [24], but may also support cancer progression in certain contexts [25]. Notably, *KLF6* could participate in T-ALL resistance by inducing iNOS, as observed in the Jurkat T-ALL cell line [26]. Additionally, *KLF6* has been implicated in the transforming activity of the oncogenic fusion protein AML-1-ETO in Acute Myeloid Leukemia (AML) [27].

Ten other genes shared at least two patients among them, suggesting that relapse could be associated with cooperative perturbations in translation (*RPs*, *EIF1*, *EEF1B2*), iron metabolism (*FTH1*), immunity (*MIF*, *EMP3*), stress responses (*UBC*, *HSP90AB1/AA1*), and metabolism (*NME1/2*, *ARL4C*).

From the 481 unique genes identified in the diagnosis and relapse modules, a sPLS-DA approach identified 41, 35, and 31 genes, respectively, for each of the three patients, showing the highest discrimination between diagnosis and relapse transcriptomes. Subsequently, we applied COX regression using transcriptomic and clinical data obtained at diagnosis from 249 T-ALL patients in the pediatric TARGET AALL0434 cohort. An AIC optimization generated model prediction signatures consisting of 11, 7, and 7 genes for each pair of diagnosis–relapse samples. Each of the three models efficiently attributed 80 to 90% of the 20 relapses in the cohort to the high-risk group, associated with a shorter patient survival as defined by the model. Notably, 60% of relapses were assigned to the high-risk group by the three models. As the cohort only provides transcriptomic data at diagnosis, we could not assess the expression of the model signatures in relapsed patients.

Within each model signature, increased expression of certain genes (*DUSP6*, *HSP90AA1*, *HSP90AB1*, *TPX2*, *TUBA1B*, *ZFP36*, *NDC80*, *ZFP36*, *MIF*) was associated with a higher relapse risk, while increased expression of others (*ACTB*, *HES4*, *HMGB2*, *JUNB*, *PHGDH*, *TUBB*, *CD52*, *DUSP2*, *MKI67*, *CDK2AP2*, *JUN*, *PFN1*, *RPS6*) was associated with a lower risk of relapse. This suggests that the expression levels and functions of the identified model genes may reveal specific biological processes or mechanisms influencing patient survival. For instance, resistance to treatments could be associated with modifications in signaling, metabolic, and proliferative pathways (*DUSP2*, *DUSP6*, *RPS6*, *PHGDH*, *MKI67*, *CDK2AP2*), a higher capacity to deal with stress responses (*HSP90AA1*, *HSP90AB1*), disruptions in gene expression programs (*HES4*, *HMGB2*, *ZFP36*, *JUN*, *JUNB*), or modifications of the cytoskeleton (*ACTB*, *NDC80*, *TUBA1B*, *TUBB*, *PFN1*). Interestingly, MIF expression has been reported as an independent prognostic factor in ALL patients [28]. These integrated bioinformatic approaches represent the initial stride towards identifying prospective therapeutic target genes.

## 5. Conclusions and Perspectives

While T-ALL is treated with chemotherapeutic drugs targeting major functions such as DNA metabolism and cell division, there is growing interest in tailored treatments for personalized medicine [4]. Dissecting cancer heterogeneity through single-cell RNAseq could support the development of precision drugs [29]. The analysis of paired diagnosis–relapse in primary T-ALL samples reveals both common and specific events during relapse. Defining the exact role of the potential relapse-associated genes will require (i) studying their role in relevant PDX models and (ii) analyzing a larger number of paired samples to cover T-ALL genetic heterogeneity [5], which likely favors the existence of different modes of resistance.

## Figures and Tables

**Figure 1 cancers-16-01667-f001:**
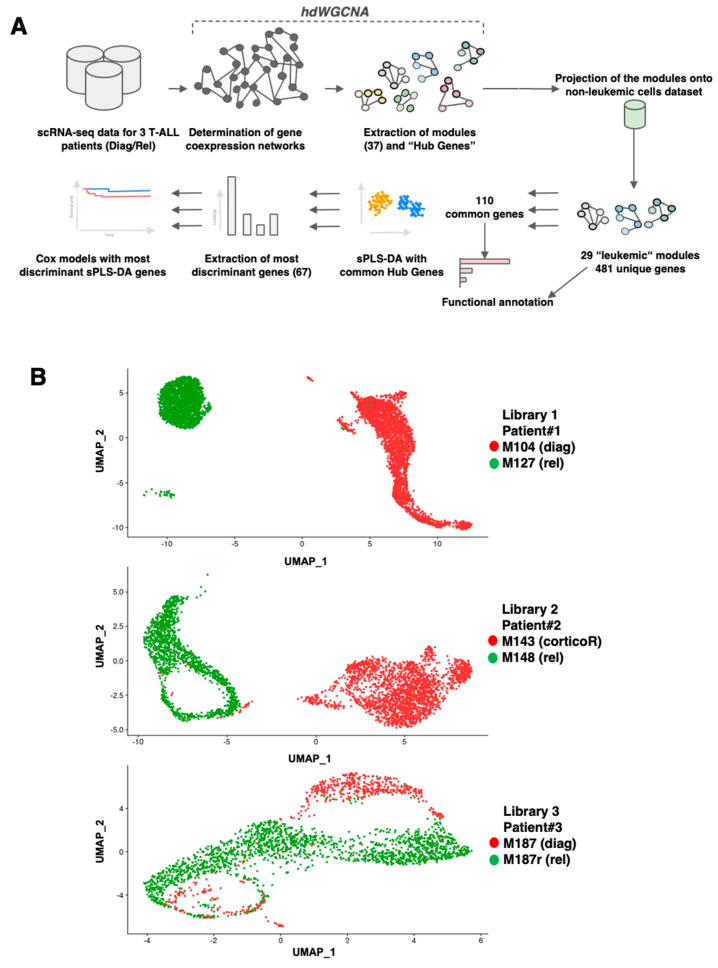
Analysis workflow and presentation of the paired T-ALL samples. (**A**) Schematic workflow of the analysis strategy. (**B**) UMAP graph of the different scRNAseq cells for the three paired samples at either diagnosis (diag), diagnosis with corticoïd resistance (corticoR), or relapse (rel). Different modules have different colors. The colors are not conserved between diagnosis and relapse states.

**Figure 2 cancers-16-01667-f002:**
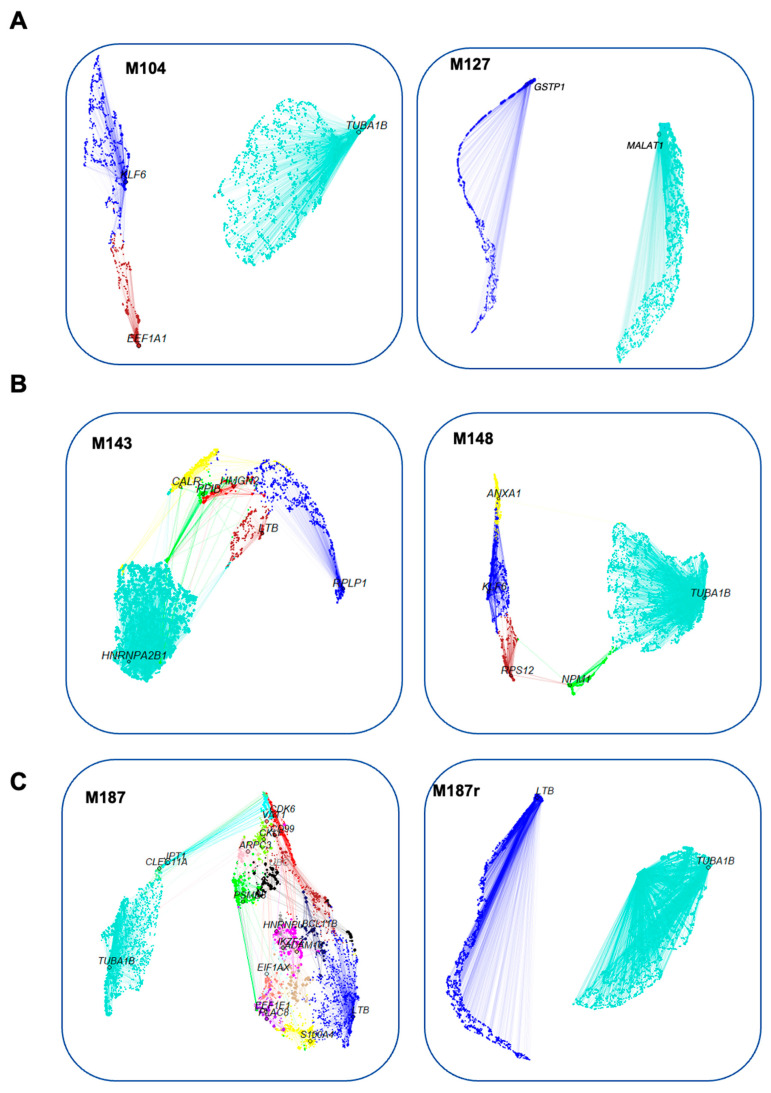
UMAP representation of gene co-expression networks. Nodes are genes, while edges are co-expression relationships between genes and hub genes of each module. The node size is scaled by kMEs. Plotted with hdWGCNA. Different modules have different colors. The colors are not conserved between diagnosis and relapse states. Subfigures represent the different samples: diagnosis samples (**left**) vs. relapse samples (**right**).

**Figure 3 cancers-16-01667-f003:**
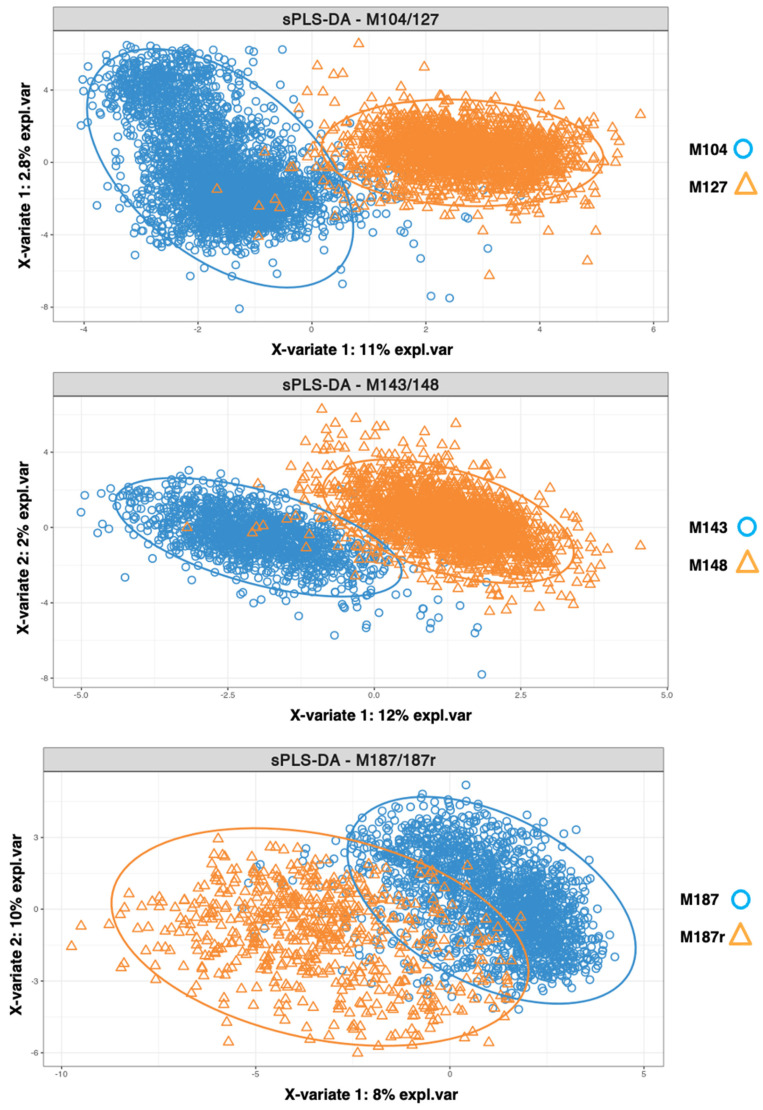
sPLS-DA analysis. The analysis was performed on the top 25 hub genes of the modules to extract the genes with the highest discriminative potential between diagnosis and relapse cells. The pair of samples for each patient is shown by a biplot.

**Figure 4 cancers-16-01667-f004:**
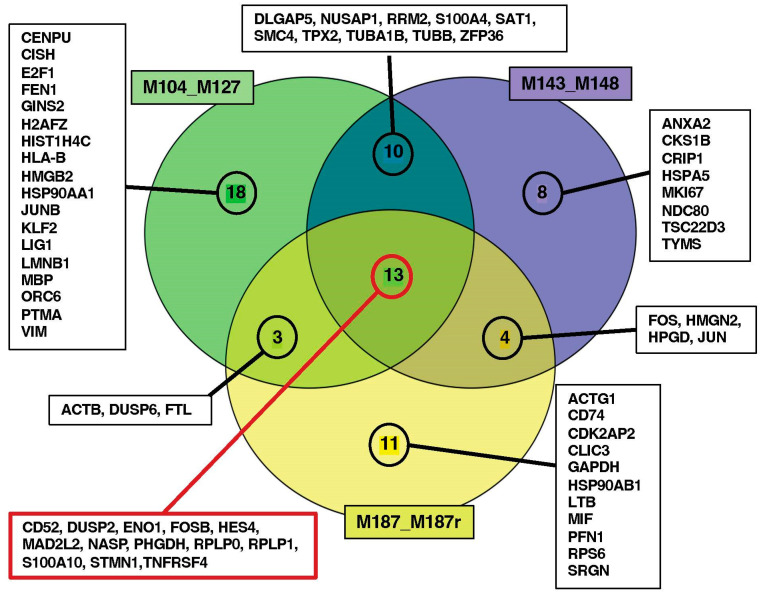
Venn diagram to illustrate the number of genes in common between the three libraries, recovered from sPLS-DA. The diagram was made with the VennDiagram package on R. The red cartouche highlight the 13 common genes.

**Figure 5 cancers-16-01667-f005:**
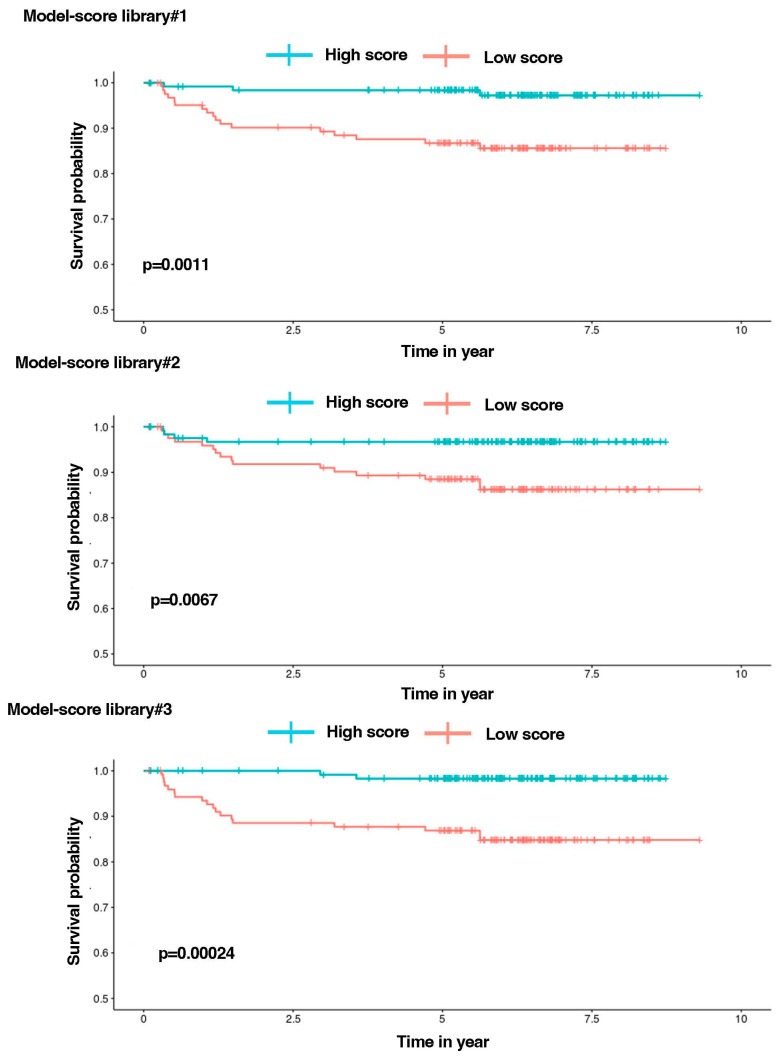
Survival analysis. Kaplan-Meier curves separating patients based on the median survival score, calculated using various coefficients determined by Cox regressions optimized by the Akaike criterion implemented for each patient pair.

## Data Availability

The single-cell datasets generated for this study were deposited on the Gene Expression Omnibus portal under the accession number GSE262272. The R scripts used in this article can be found at Peyronlab.github.io.

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
