# Peer review of "Identifying Candidate Gene Drivers Associated with Relapse in Pediatric T-Cell Acute Lymphoblastic Leukemia Using a Gene Co-Expression Network Approach"

_cancers, 2024, doi:10.3390/cancers16091667_

Round 1

Reviewer 1 Report

Comments and Suggestions for Authors

This is an interesting paper in which the authors use scRNAseq data from 3 T-ALL paediatric patients with diagnosis-relapse paired-samples. From these data, they use computational methods to construct gene coexpression networks (characterized by genes with similar expression trends), in which they select 37 gene modules characterized by differential expression between diagnosis and relapse for each patient. From them, they select 29 gene modules that are specific of leukemic (and not normal) cells. They propose that these 29 gene modules are drivers of relapse, and thus that 1) they can inform about the mechanisms involved in the development of relapse in T-ALL patients; and 2) they can help to risk-stratify patients. In each of the 29 gene modules, they select 25 hub genes (those with the highest connectivity within their module). They obtain a list of 481 unique genes. If I understood correctly, from them they select 110 genes that overlap between the three patients; these will finally constitute the best predictors of relapse. Next, they select 67 unique genes, from which 44, 35 and 31 genes are highly discriminating for patients #1, #2 and #3, respectively, and 13 genes are common for the three patients. From these three lists of genes (with 44, 35 and 31), they develop a scoring system whereby they multiply the expression of each gene (taken from RNAseq data from the TARGET cohort, n=249 patients) by its corresponding Cox coefficient (taken from the survival analysis in the same cohort). This way, they associate the expression with the survival in order to make three models with the best performance. Using these three models, they classify in the high-risk groups 17/20, 16/20 and 18/20 patients with relapse, respectively.

Major comments:

- Results, line 211 (“3.3. Search for potential conserved driver genes”): The authors hypothesize that these common genes could be leukemic drivers. However, it cannot be ruled out that they are passengers. Functional studies would be needed to propose them as drivers, so it would be probably better to just discuss this matter in the Discussion section.

- Results, lines 233-234: The authors indicate that they “extracted the 10 variables with the largest loadings per component, most involved in the diagnosis / relapse discrimination”. I do not understand what that of “the largest loadings per component” means; for general readers not so expert in bioinformatics, I think it would be useful to further explain this concept in the manuscript.

- Results, line 241 (“3.5. Establishment of library gene signatures and correlation with patient’s survival”): With this approach, the authors associate the expression of selected genes with patients´ survival in the TARGET cohort, in order to make three models with the best performance. Using these three models, they classify in the high-risk groups 17/20, 16/20 and 18/20 patients with relapse, respectively. My questions are the following:

1) If I understood right, these genes arise from a previous list that was based on differential expression between diagnosis and relapse. However, the RNAseq data from TARGET cohort are limited to the diagnosis samples, and there is no expression data from the 20 relapses. I understand then that the authors do not test the discriminatory validity of these genes between Dx and Rx, but only use their expression at Dx to assign patients a risk score (high vs. low) and test if indeed the expression of the gene signature at Dx predicts relapse. This limitation should be commented in the manuscript.

2) There are 3, 4 and 2 patients with Rx who do not fall into the high-risk groups for models 1, 2 and 3, respectively. Do they then fall into the low-risk group? This should be indicated.

3) Conversely, what about the patients who do not relapse (the majority in the cohort), do some of them fall into the high-risk group with any of the 3 models? This should be mentioned in the Results and/or Discussion sections.

- Discussion, lines 334-336: In my opinion, it would be necessary to discuss all this a little more, to interpret precisely what these biological processes and mechanisms involved would be, according to the signature; and also to discuss whether there are more data in the literature that attribute prognostic or predictive risk value to any of these genes.

Minor comments:

- Supplementary table 1E: label for patient#3 is missing

- Resolution of figures 1B, 2, 5, … is very poor, labels cannot be read.

- Supplemental figures are labeled in a confusing way: S3 is renamed as Supplemental Figure 2bis, there is “Fig.4” and “Fig. Sup. 4” in the same powerpoint file for supplementary figure; supplemental figure 6 is mentioned at the end of the Results section, but it is not present in this Supplemental material.

Comments on the Quality of English Language

English language is fine, just with minor mistakes that should be revised.

Author Response

>We sincerely thank the reviewer for the constructive comments. We tried to address all the concerns and we hope we bring satisfactory answers. We consider that the quality of our manuscript was greatly improved by the corrections we made. Thank you for your work.

Reviewer#1

Open Review

Comments and Suggestions for Authors

This is an interesting paper in which the authors use scRNAseq data from 3 T-ALL paediatric patients with diagnosis-relapse paired-samples. From these data, they use computational methods to construct gene coexpression networks (characterized by genes with similar expression trends), in which they select 37 gene modules characterized by differential expression between diagnosis and relapse for each patient. From them, they select 29 gene modules that are specific of leukemic (and not normal) cells. They propose that these 29 gene modules are drivers of relapse, and thus that 1) they can inform about the mechanisms involved in the development of relapse in T-ALL patients; and 2) they can help to risk-stratify patients. In each of the 29 gene modules, they select 25 hub genes (those with the highest connectivity within their module). They obtain a list of 481 unique genes. If I understood correctly, from them they select 110 genes that overlap between the three patients; these will finally constitute the best predictors of relapse. Next, they select 67 unique genes, from which 44, 35 and 31 genes are highly discriminating for patients #1, #2 and #3, respectively, and 13 genes are common for the three patients. From these three lists of genes (with 44, 35 and 31), they develop a scoring system whereby they multiply the expression of each gene (taken from RNAseq data from the TARGET cohort, n=249 patients) by its corresponding Cox coefficient (taken from the survival analysis in the same cohort). This way, they associate the expression with the survival in order to make three models with the best performance. Using these three models, they classify in the high-risk groups 17/20, 16/20 and 18/20 patients with relapse, respectively.

>This is a very nice and comprehensive summary of our approach and results by reviewer#1

Major comments:

- Results, line 211 (“3.3. Search for potential conserved driver genes”): The authors hypothesize that these common genes could be leukemic drivers. However, it cannot be ruled out that they are passengers. Functional studies would be needed to propose them as drivers, so it would be probably better to just discuss this matter in the Discussion section.

>We agree with the reviewer. WGCNA allows the identification of hub genes which are the highest connected genes within modules. Because of that, variations in the expression of these genes will affect numerous connected genes with the highest impact on the studied biological process. In a normal physiological situation, for instance cell differentiation, hub genes have a high probability to be the drivers of the process. We agree that in cancer cells, the situation might be less monolithic. Cancer cells are associated with an important replicative stress and genomic instability, that create a noisy background of gene dysregulation. Therefore, we agree that without adequate functional studies, we cannot claim that the hub genes we identified are relapse driver genes. This is why we constantly used the adjective POTENTIAL gene drivers in the manuscript. Nevertheless, as suggested, we changed the title of the 3.3 paragraph from: “Search for potential conserved driver genes“ to “Search for conserved relapse-associated hub genes“. We replaced the line 211 sentence by “We compared the gene networks for each library to select the most conserved hub genes.  and discussed about hub/driver genes at the beginning of the discussion:

We focused on relapse-associated gene networks and hub genes and their conservation between several patients. Hub genes that correspond to the most connected genes within their module, are envisioned as the regulators of the module. In our case, hub genes from relapse-specific modules could be considered as potential drivers or at least important players of relapse. Further biological experiments will be nevertheless required to evaluate their real driver potential.  (lanes 325-330 of the revised-marked manuscript).

- Results, lines 233-234: The authors indicate that they “extracted the 10 variables with the largest loadings per component, most involved in the diagnosis / relapse discrimination”. I do not understand what that of “the largest loadings per component” means; for general readers not so expert in bioinformatics, I think it would be useful to further explain this concept in the manuscript.

>We understand the concern of the reviewer. We thus simplified the text in the Result section as follows: “The statistical method was used to highlight the most important genes that distinguish the diagnostic and relapse status. It identified 67 unique genes distributed as 44, 35 and 31 highly discriminating genes for respectively patients #1, #2 and #3  (lanes 280-288 of the revised-marked manuscript).

In addition, the description of sPLS-DA has been simplified as follows:

The top 25 hub genes for each module were selected according to their kME (Module Eigengene-based connectivity) value that provides a quantitative measure of how closely their gene expression pattern aligns with the overall expression pattern of a module (1st Component). 

We next applied a sparse partial least square discriminant analysis (sPLS-DA) (R MixOmics package) to identify the most discriminant genes between diagnostic and relapse [20]. During sPLS-DA, each variable's contribution to the discrimination between classes is measured through loadings. Loadings represent the direct regression coefficients between the original variables and the latent components (linear combinations of the original variables) extracted by the algorithm. The variables with the largest loadings have the most influence in determining the components (Figure S4).

To maximize sPLS-DA discrimination capacity, the number of components was chosen with the MixOmics perf function with 10 folds and 50 repetitions, and the tune function used to define the number of variables to be considered by components, with 10 folds and 100 repetitions applied to different vectors assigning each component with different numbers of variables. 

A vector is defined between a low and a high limit, with a resolution serving to narrow the interval between them. On each iteration the resolution is increased based on the output of the tune function.

Once each patient’s library had been processed, the genes with the highest discriminative power were extracted. For each loading vector, we recovered the 10 top genes (or all the vectors if the components contained less than 10 genes) with the best loading vector absolute values per component. 

(lanes 182-209 of the revised-marked manuscript).

- Results, line 241 (“3.5. Establishment of library gene signatures and correlation with patient’s survival”): With this approach, the authors associate the expression of selected genes with patients´ survival in the TARGET cohort, in order to make three models with the best performance. Using these three models, they classify in the high-risk groups 17/20, 16/20 and 18/20 patients with relapse, respectively. My questions are the following:

1) If I understood right, these genes arise from a previous list that was based on differential expression between diagnosis and relapse. However, the RNAseq data from TARGET cohort are limited to the diagnosis samples, and there is no expression data from the 20 relapses. I understand then that the authors do not test the discriminatory validity of these genes between Dx and Rx, but only use their expression at Dx to assign patients a risk score (high vs. low) and test if indeed the expression of the gene signature at Dx predicts relapse. This limitation should be commented in the manuscript.

>Exactly. Unfortunately, we could not test the discriminatory potential of our genes in the Target cohort as no RNAseq data are available for Rx patients. This limitation is now mentioned in the Discussion:

“As the cohort only provides transcriptomic data at diagnostic, we could not assess the expression of our model signatures to relapsed patients.“ (lanes 390-392 of the revised-marked manuscript).

2) There are 3, 4 and 2 patients with Rx who do not fall into the high-risk groups for models 1, 2 and 3, respectively. Do they then fall into the low-risk group? This should be indicated.

>Reading this important question raised by the reviewer, we realized that we introduced a confusion in the manuscript between the model score and the patient’s risk. We classified patients in the low/high groups according to their individual scores, respectively lower or higher than the mean of the scores of the cohort. If a high score is clearly associated with a lower survival of the patients, we cannot measure its eventual correlation with a high-risk classification for the patient as the cohort does not provide any indication of the risk classification (intermediate/high) of the patients.

The mean of the cohort splits the patients in approximatively a 50-50 distribution which is different from a 60-40 distribution for the true proportion of intermediate/high risk patients. It is therefore not surprising that some patients who did not relapse were classified in the high score group, which again is different from the high-risk classification. In the reverse situation, also 50% of patients who did not relapse fall into the high score group.

This shows that the high scores of the patients are associated with a shorter survival of the patients, suggesting that the three gene signatures contain genes involved with leukemia progression. However, the model signatures cannot be used to predict the relapse risk.

The text has therefore been modified to remove this misunderstanding as follows:

Abstract:

“Overall, our approach identified new potential relapse-associated driver genes and biomarkers and propose three model gene signatures associated with high-risk patients.“

Changed by:

“Overall, our approach identified new potential relapse-associated driver hub genes and biomarkers and propose three model gene signatures associated with a lower survival for the high-score patients high-risk patients.“

Introduction:

…we validated three gene signatures that were able, at initial diagnosis, to assign the patients to a high-risk group, independently from the response to therapy, as measured by post-induction minimal residual disease.

Changed by:

…we validated three model gene signatures that using transcriptomic data obtained at initial diagnosis could assign 85% of the 20 relapsed patients into a low survival group.

group, independently from the response to therapy, as measured by post-induction minimal residual disease.

Results:

Each model score could respectively classify 85% (17 patients), 80% (16) and 90% (18) of the 20 relapses of the cohort in the "High " group.

Changed by:

Each model score could respectively classify 85% (17 patients), 80% (16) and 90% (18) of the 20 relapses of the cohort in the High-score group. Besides, 47% of the patients who did not relapse were attributed a high-score.

Discussion:

Each of the three models could efficiently attribute 80 to 90% of the 20 relapses of the cohort, to the model-defined high-risk group associated with a shorter survival. Importantly, 60% of relapses were assigned to the high-risk group by the 3 models.

Changed by:

Each of the three models could efficiently attribute 80 to 90% of the 20 relapses of the cohort, to the model-defined high-score group associated with a shorter survival. Importantly, 60% of relapses were assigned to the high-risk group by the 3 models, suggesting an involvement of the model genes during leukemia evolution. However, the score cannot be used to anticipate the risk of relapse for the patients.

Within each model signature, the increase expression of some genes (DUSP6, HSP90AA1, HSP90AB1, TPX2, TUBA1B, ZFP36, NDC80, ZFP36, MIF) was associated with a higher relapse risk. Alternatively, the increased expression of others (ACTB, HES4, HMGB2, JUNB, PHGDH, TUBB, CD52, DUSP2, MKI67, CDK2AP2, JUN, PFN1, RPS6) was associated with a lower risk of relapse.

Changed by:

Within each model signature, the increase expression of some genes (DUSP6, HSP90AA1, HSP90AB1, TPX2, TUBA1B, ZFP36, NDC80, ZFP36, MIF) was associated with a higher score. Alternatively, the increased expression of others (ACTB, HES4, HMGB2, JUNB, PHGDH, TUBB, CD52, DUSP2, MKI67, CDK2AP2, JUN, PFN1, RPS6) was associated with a lower score.

3) Conversely, what about the patients who do not relapse (the majority in the cohort), do some of them fall into the high-risk group with any of the 3 models? This should be mentioned in the Results and/or Discussion sections.

See our answer to question#2

- Discussion, lines 334-336: In my opinion, it would be necessary to discuss all this a little more, to interpret precisely what these biological processes and mechanisms involved would be, according to the signature; and also to discuss whether there are more data in the literature that attribute prognostic or predictive risk value to any of these genes.

>We followed the reviewer’s recommendation and added to the discussion some comments concerning the genes associated with the model scores.

Minor comments:

- Supplementary table 1E: label for patient#3 is missing

>corrected

- Resolution of figures 1B, 2, 5, … is very poor, labels cannot be read.

>We apologize for this problem. We increased the quality of the figures in the revised manuscript.

- Supplemental figures are labeled in a confusing way: S3 is renamed as Supplemental Figure 2bis, there is “Fig.4” and “Fig. Sup. 4” in the same powerpoint file for supplementary figure; supplemental figure 6 is mentioned at the end of the Results section, but it is not present in this Supplemental material.

>We also apologize for this lack of consistency in naming the figures. This problem was corrected.

Comments on the Quality of English Language

English language is fine, just with minor mistakes that should be revised.

>We revised the manuscript to correct the mistakes mentioned by both reviewers.

Reviewer 2 Report

Comments and Suggestions for Authors

This is a commendable work from Kypraios et al. about pediatric T-ALL, which brings new crucial single-cell RNA-seq data to the table with matched samples of patients at diagnosis and relapse. This topic and data is of interest to the scientific community. The methods used are standard single-cell RNA sequencing performed with a standard and well known platform such as 10x, with complex statistical analysis, intending to elucidate modules of genes that are specific to the relapse. The overall goal is to provide help in identifying high risk patients prone to relapse, an unmet need in the field today, which highlights the importance of this and similar works.

Despite the importance of the data and results obtained and described by the authors, I have a series of major and minor concerns that, in my opinion, severly reduce the quality and potential impact of this publication.

First and foremost, the quality/resolution of the figures provided with the manuscript is too low, resulting in generally hard to interpret figures. Legends are not visible, sometimes overlapped with added text on top. I suggest to properly format figures, legends, and describe more thoroughly and extensively what is represented in the figure legend text.

Another major concern is that the more basic methods and technical results of the single-cell are just insufficient. I suggest to describe more extensively all the steps for the single-cell, including alignment, specifiying programs and their version. I would also pose more attention in describing the QC metrics during sequencing, and specifying details such as targeted reads per cells, etc.

Following up on the previous point, there is a lack of information about how many cells were obtained in the end, how many remained after QC such as doublet removal and cells with high mitochondrial reads removal. Plots related to this, and genes per cell, could be included in supplementary to give the reader a clearer picture. Generally, I would suggest describe more extensively the methods and QC results.

It seems that more basic single-cell analysis was not considered for this work. It would still be of interest to the readers to have a general analysis by cluster, to understand the structure of the three different patients and have a better idea of the general landscape being described. This could also be done by separating the samples at diagnosis from that at relapse.

The analysis, overall, is more technical and specific. I would suggest to the authors to consider providing the code or the workflow of the analysis. This would help in re-analyzing and evaluate the results, and would be beneficial in making workflows like this more available and accesible to the scientific community, increasing the overall impact of the work.

MALAT1 is often being detected in poly-A capture strategies related to single-cell, and correlates with cell death or general health. Some analysts even consider removing it from their analysis. Did the author took any extra precautions before reporting this gene, as it could be influenced by factor more independent of the  pathological biology of the cells? I acknowledge and appreciate the extensive part about MALAT1 in the discussion section, but I would also like to see more discussion to exclude that MALAT1 is there more due to technical reason, and that it has to do with the pathobiology of T-ALL.

In figure 2, the authors report a UMAP plot for gene coespression networks. For this network, did the author consider something like force-directed layout such as ForceAtlas? Would this improve the visualization?

The authors report in the table the molecular groups of the T-ALL samples. At the level of pseudobulk, how they sample compare to those reported by Mullighan in the TARGET cohort? Can the authors elaborate more on this?

in Method 2.5, the authors described their use of healthy T-cells from a 10x dataset. Considering that the authors acknowledge in their introduction that T-ALL is related to block of differentiation of T-ALL progenitors, would it not make more sense to compare with thymic precursors specializing towards T-cells? Can the authors elaborate more on this? A great single-cell dataset was published in a recent science paper by Park et al. in 2020, PMID: 32079746.

In a recent paper published on Blood in 2023, the group of Dr. Giambra (PMID: 36315912) utilized a similar dataset, with 3 patients at diagnosis and relapse. Considering the scarcity of such datasets, did the author consider adding these as a comparison in their study? What the authors think about potentially doubling the sample size? Could this strengthen and improve their findings, also extended to a large cohort such as the TARGET one, considering the variability and molecular subgroups reported there for T-ALL? 

Comments on the Quality of English Language

There are some minor corrections to be made. Readability and fluency can be improved. As an example of some corrections to be made, see lines:

42

47-48

81

92-93

150

Figure A, legend text, last line

200

330

Author Response

>We sincerely thank the reviewer for the constructive comments. We tried to address all the concerns and we hope we bring satisfactory answers. We consider that the quality of our manuscript was greatly improved by the corrections we made. Thank you for your work.

Reviewer#2

Open Review

Comments and Suggestions for Authors

This is a commendable work from Kypraios et al. about pediatric T-ALL, which brings new crucial single-cell RNA-seq data to the table with matched samples of patients at diagnosis and relapse. This topic and data is of interest to the scientific community. The methods used are standard single-cell RNA sequencing performed with a standard and well known platform such as 10x, with complex statistical analysis, intending to elucidate modules of genes that are specific to the relapse. The overall goal is to provide help in identifying high risk patients prone to relapse, an unmet need in the field today, which highlights the importance of this and similar works.

Despite the importance of the data and results obtained and described by the authors, I have a series of major and minor concerns that, in my opinion, severly reduce the quality and potential impact of this publication.

First and foremost, the quality/resolution of the figures provided with the manuscript is too low, resulting in generally hard to interpret figures. Legends are not visible, sometimes overlapped with added text on top. I suggest to properly format figures, legends, and describe more thoroughly and extensively what is represented in the figure legend text.

>We really apologize for the poor quality of the figures. All figures, legends were reviewed and modified in the revised manuscript.

Another major concern is that the more basic methods and technical results of the single-cell are just insufficient. I suggest to describe more extensively all the steps for the single-cell, including alignment, specifiying programs and their version. I would also pose more attention in describing the QC metrics during sequencing, and specifying details such as targeted reads per cells, etc.

Following up on the previous point, there is a lack of information about how many cells were obtained in the end, how many remained after QC such as doublet removal and cells with high mitochondrial reads removal. Plots related to this, and genes per cell, could be included in supplementary to give the reader a clearer picture. Generally, I would suggest describe more extensively the methods and QC results.

>We agree with the reviewer. We did not realize that we downplayed the scRNAseq approach and took it for granted. We now provide the GC metrics (Table S1B) as well as a GSE access number. 

It seems that more basic single-cell analysis was not considered for this work. It would still be of interest to the readers to have a general analysis by cluster, to understand the structure of the three different patients and have a better idea of the general landscape being described. This could also be done by separating the samples at diagnosis from that at relapse.

The analysis, overall, is more technical and specific. I would suggest to the authors to consider providing the code or the workflow of the analysis. This would help in re-analyzing and evaluate the results, and would be beneficial in making workflows like this more available and accesible to the scientific community, increasing the overall impact of the work.

>We thank the reviewer for these suggestions. We provide (Figure S1) an UMAP representation of the 6 samples separating the diagnosis and the relapse. This figure shows the interpatient heterogeneity as well as an intra patient heterogeneity between diagnosis and relapses states. However, we do not provide any gene analysis of the different clusters. This is an ongoing study to understand gene plasticity and adaptation in the evolution to relapse.

As suggested, we provide a link to a GitHub for the readers to have access to the different script in R that we used: hdWGCNA, hdWGCNA healthy cells projection, sPLS-DA, coxph_optimization, ForceAtlas.

MALAT1 is often being detected in poly-A capture strategies related to single-cell, and correlates with cell death or general health. Some analysts even consider removing it from their analysis. Did the author took any extra precautions before reporting this gene, as it could be influenced by factor more independent of the  pathological biology of the cells? I acknowledge and appreciate the extensive part about MALAT1 in the discussion section, but I would also like to see more discussion to exclude that MALAT1 is there more due to technical reason, and that it has to do with the pathobiology of T-ALL.

>We really thank the reviewer for this helpful comment, as we were not aware of this potential problem with MALAT1. Indeed, it appears that MALAT1 has a long internal AAAA stretch that could make it over-represented in the analysis. As wisely suggested by the reviewer, we decided to delete the text concerning MALAT1.

In figure 2, the authors report a UMAP plot for gene coespression networks. For this network, did the author consider something like force-directed layout such as ForceAtlas? Would this improve the visualization?
>We followed the reviewer’s advice and performed a ForceAtlas representation of the gene-coexpression networks. It is displayed on Figure S6. The ForceAtlas representation shows the relationships between modules while the UMAP representation also highlights the relationships between genes within their modules.

The authors report in the table the molecular groups of the T-ALL samples. At the level of pseudobulk, how they sample compare to those reported by Mullighan in the TARGET cohort? Can the authors elaborate more on this?

>It is an interesting suggestion to use a pseudobulk approach to compare our patients with those from the Mullighan’s cohort. However, we think that the relapse events are likely to be highly diverse from one patient to another, because T-ALL is highly heterogeneous at a genetic level and because of the differences in response to treatments for the patients. So, even if we have a better identification of our samples, we think that we will not be able to extend our findings to samples of the same T-ALL subgroup.

We believe that the strength of our analysis is to use paired samples from the same patient to have a more authentic access to relapse mechanisms. But we are aware that having only 1 or 2 patients of one T-ALL subgroup is a strong limitation if we want to draw wider conclusions.

In addition to the relapse-associated hub genes identified here, we are currently searching for Dx/Rx differentially expressed genes in the 3 libraries. From our preliminary results it seems that the relapse mechanisms may be mostly patient-specific rather than subgroup-specific. With these 2 approaches we hope to be able to define subsets of patients with specific gene/hub genes signatures within the 3 main T-ALL molecular subgroups. We think the situation could be similar to that of DLBCL. Indeed, 20 years ago DLBCL was first transcriptionally defined in 2 groups ABC/GCB, but is now composed of at least 6 genetic subtypes.

We will use the pseudobulk approach anyway in our follow-up study.

in Method 2.5, the authors described their use of healthy T-cells from a 10x dataset. Considering that the authors acknowledge in their introduction that T-ALL is related to block of differentiation of T-ALL progenitors, would it not make more sense to compare with thymic precursors specializing towards T-cells? Can the authors elaborate more on this? A great single-cell dataset was published in a recent science paper by Park et al. in 2020, PMID: 32079746.

>The differentiation blockage associated with leukemogenesis may occur at different stages in different patients.  Considering the use of datasets from T-cell progenitors as healthy T cell control is a valid suggestion by the reviewer. Nevertheless, it might overlook some aspects of the disease biology present in more differentiated T-cell populations, meaning that the perfect healthy control may not exist. So, we consider it would be a lot of work to start the analysis with a new control group to end up with similar or different gene modules and hub genes. But we agree it is a legitimate suggestion.

In a recent paper published on Blood in 2023, the group of Dr. Giambra (PMID: 36315912) utilized a similar dataset, with 3 patients at diagnosis and relapse. Considering the scarcity of such datasets, did the author consider adding these as a comparison in their study? What the authors think about potentially doubling the sample size? Could this strengthen and improve their findings, also extended to a large cohort such as the TARGET one, considering the variability and molecular subgroups reported there for T-ALL? 

Besides this article, we are currently analyzing, on the 3 patient pairs, the leukemic clonal evolution between the diagnostic and relapse states. Again, our goal is to see if some of these relapse-associated genes/hub genes can be observed in other patients, to identify more precise subsets. As the number of transcriptomic reports on Dx-Rx paired T-ALL patients are rare, the possibility to use the data from Dr Giambra’s group is interesting and valuable. Thank you. We will contact this group. In another collaboration, we have access to several other Dx-Rx T-ALL pairs that will be also explored. 

Comments on the Quality of English Language

There are some minor corrections to be made. Readability and fluency can be improved. As an example of some corrections to be made, see lines:

>We carefully read our manuscript to improve readability and fluency.

42 >Corrected: corticoids instead of steroids

47-48 >Corrected: It is thus crucial to identify

81 >Corrected: 2.2 10XGenomics

92-93 >Corrected: Selected cells displayed i)….

150 >The 2.7 paragraph “Most discriminating genes between the two states“ was reorganized and simplified.

Figure A, legend text, last line >The figure legends were re-written

200 >Corrected: relapse

330 >Corrected: increased

Round 2

Reviewer 2 Report

Comments and Suggestions for Authors

I appreciate the effort of the authors in addressing my concerns. The methods are now exhaustive and the results are easier to follow and more sound.

I also appreciate the authors sharing the code and data.

Overall, there are some minor edits to the English and other minor imperfections that need to be done. For example in Figure 4, all the genes are shown in a table, but for the overlap showing 13 genes, only 10 are reported in the table.

Aside from this, my opinion is that the manuscript is now suitable for publication on Cancers.

Comments on the Quality of English Language

Editing and proofreading is required.

Author Response

As noticed by the reviewer, 3 genes were missing on the overlap of Figure 4. They have been added back. We provide the revised figure 4.

We also edited the text to improve the style and the english grammar. We provide the revised 2 version of the manuscript. The editing includes a slightly different title.

We added new authors to the manuscript and provide the Change in Authorship form.

Thanks
